# Uncommon Association of Mckittrick-Wheelock Syndrome and *Clostridioides difficile* Infection in Acute Renal Failure

**DOI:** 10.3390/diagnostics12040784

**Published:** 2022-03-23

**Authors:** Irina Ciortescu, Vasile-Liviu Drug, Oana-Bogdana Bărboi, Denis Pleșca, Roxana Livadariu, Lidia Ionescu

**Affiliations:** 1Medical I Department, “Grigore T. Popa” University of Medicine and Pharmacy, 700115 Iasi, Romania; irinaciortescu@yahoo.com (I.C.); vasidrug@email.com (V.-L.D.); 2Institute of Gastroenterology and Hepatology, Saint Spiridon Hospital, 700111 Iasi, Romania; plescad@gmail.com; 31st Surgery Department, “Grigore T. Popa” University of Medicine and Pharmacy, 700115 Iasi, Romania; roxanalivadariu@yahoo.com (R.L.); lidia.ionescu07@yahoo.com (L.I.); 43rd Surgery Department, Saint Spiridon Hospital, 700115 Iasi, Romania

**Keywords:** *Clostridioides difficile*, McKittrick-Wheelock syndrome, acute renal failure

## Abstract

We present the case of a 71-year-old male who suffered an episode of acute renal failure caused by the uncommon association of two different diseases (*Clostridioides difficile* infection and McKittrick-Wheelock syndrome). He presented with hypovolemic shock, severe hypokalemia, hyponatremia, metabolic acidosis and acute renal failure; consequences of secretory diarrhea caused by a giant rectal tumor revealed from colonoscopy. The biopsy results revealed tubulo-villous adenoma with low/high grade dysplasia. After correction of electrolyte imbalances and azotemia, the patient underwent surgical resection with full subsequent recovery. In the literature review, including papers published from which January 1945 to April 2021, we found only one case-report of acute renal failure associated with *Clostridioides difficile* infection and McKittrick-Wheelock syndrome.

## 1. Introduction

McKittrick-Wheelock syndrome was first described in 1954 [1]. The syndrome is caused by a giant distal colorectal tumor, usually benign villous adenomas, which secrete high quantities of electrolyte-rich mucin. Patients usually present with secretory mucous diarrhea, dehydration, acute kidney injury, severe hypokalemia and hyponatremia.

*Clostridioides difficile*-associated diarrhea is currently the most frequent nosocomial infection in many hospitals in Europe and shows increasing rates in North America [2,3]. There is also an increase in the frequency and severity of community acquired *Clostridioides difficile* infection in the population not previously considered to be at risk. The population groups particularly at risk are the elderly, immunocompromised patients and those with recent antibiotic exposure, although studies show that only 37% of patients received antibiotic therapy in the 90 days prior to the diagnosis [2,4]. In this paper, we present the case of a patient who suffered an episode of acute renal failure, hyponatremia, hypokalemia and metabolic acidosis, without any contact with a healthcare facility in the previous 12 weeks.

## 2. Case Presentation

A 71-year-old Caucasian male presented to our emergency department with nausea, vomiting and abundant loose stools, starting 2 weeks before, and anuria with the onset 2 days prior to this admission. His medical history documented insulin-dependent type 2 diabetes for 20 years with microangiopathic complications. He had no history of recent hospitalization or other medication.

Clinical examination evaluated a lean male (BMI = 23.1 kg/m^2^) who was clinically dehydrated, tachycardic (115 bpm) and hypertensive (140/70 mmHg). Blood testing revealed severe renal failure with blood urea = 300 mg/dL, serum creatinine = 11 mg/dL, HCO_3_ = 14 mmol/L, serum Na = 130 mmol/L, serum K = 2.8 mmol/L (Table 1). Prerenal failure was assumed as the primary working diagnosis as a consequence of dehydration due to prolonged diarrhea. Considering the diagnosis and hemodynamic instability, the patient was hospitalized in the Nephrology Department, where the patient was treated with intravenous fluids and electrolytes and one hemodialysis session. Stool testing revealed the presence of *Clostridioides difficile* GDH, toxin A and toxin B by direct enzyme immunoassay (Table 2). Treatment with oral vancomycin and intravenous metronidazole was started. The patient denied any recent exposure to antibiotics or any contact with a healthcare facility in the previous 12 weeks. Therefore, community acquired *Clostridioides difficile* infection was considered.

Whilst the patient was hospitalized in the Nephrology Department, there was a considerable decrease in blood urea and creatinine values, but the diarrheal syndrome persisted (10–15 watery stools daily). Because of this persisting symptom, the patient was transferred to the Gastroenterology Department for further investigations.

On admission, the patient was clinically dehydrated, tachycardic (95 bpm) and normotensive (110/75 mmHg). Blood testing revealed blood urea = 110 mg/dL and serum creatinine = 1.4 mg/dL, HCO_3_ = 24.5 mmol/L, serum Na = 135 mmol/L, K = 3.4 mmol/L (Table 1). Digital rectal examination revealed a soft spongy mass at fingertip.

Considering the patient’s profuse diarrhea with 10–12 watery or mucous stools/24 h (Figure 1), a colonoscopy was performed. The examination revealed 12 cm length circumferential lesion located at 8 cm from the anal verge (Figure 2). Histology revealed tubulo-villous adenoma with low-grade dysplasia. Another villous adenoma measuring 2/2 cm was identified on the descending colon, before the splenic flexure, histology revealing tubulo-villous adenoma with high-grade dysplasia. In addition, a semi-pedunculated polyp, 1 cm in diameter (Isp on Paris classification), was found on the right colon, 4 cm from the ileocecal valve.

Thoracic, abdominal and pelvic computerized tomography (CT) scan was performed, revealing a large and circumferential mass of 12 cm in length, located 8 cm from the anal verge. The recto-sigmoid wall was 2 cm in thickness, with no evidence of extramural invasion (Figure 3).

While the patient was hospitalized in our department, subsequent stool samples showed the persistence of *Clostridioides difficile* toxins despite the two lines of treatment administered successively (oral Vancomycin plus Metronidazole and then Fidaxomicin) (Table 2).

Regarding treatment, volume and electrolyte replacement was maintained, with gradual correction of hyponatremia, hypokalemia and azotemia. For the definitive treatment the patient was transferred to the Surgical Department and successfully underwent left hemicolectomy (Hartmann procedure) (Figure 4). The histology of postoperative specimen reported tubulo-villous adenoma with low-grade dysplasia and several foci of high-grade dysplasia, without any signs of infiltration in muscularis mucosae and beyond it (Figure 5). There were also examined 12 lymph nodes, with non-specific reactive inflammatory changes. Corroborating histopathological aspects with clinical and biochemical features, the diagnosis of McKittrick-Wheelock syndrome was considered.

Following surgery, renal function and electrolytes remained normal and the patient no longer reported watery or mucous stools. One week after surgery, the patient’s stool became negative for *Clostridioides difficile.* He was discharged 10 days after surgery with favorable evolution, resulting in full recovery.

## 3. Discussion

In 1954, McKittrick and Wheelock reported a syndrome characterized by mucinous diarrhea associated with severe volume and electrolyte depletion (hyponatremia, hypokalemia) and severe dehydration and pre-renal azotemia caused by a giant villous adenoma of the rectum or sigmoid colon [1]. There are about 50 cases of McKittrick-Wheelock syndrome reported in the literature [5,6]. Villous adenomatous polyps underlying McKittrick-Wheelock syndrome are generally benign, with 15–25% malignant potential, which is even higher (about 40%) for polyps over 4 cm in diameter. Although up to two-thirds of polyps are found in the rectum, they can also be identified in the recto-sigmoid, cecum, ascending colon, stomach or duodenum [7].

Large villous adenomas of the rectum or recto-sigmoid can rarely cause secretory mucinous diarrhea, characterized by massive fluid loss, acute renal failure and electrolyte depletion, due to the distal location of the tumor that does not allow adequate reabsorption [7,8]. Adenoma must measure between 7–18 cm in diameter and cover a large percentage of the mucosal surface of the rectum or colon in order to determine significant volume loss [9,10]. Studies showed that in patients with villous adenomas of the rectum, the levels of prostaglandin E2 (PGE2) and intracellular cyclic adenosine monophosphate (cAMP) are higher in rectal secretions. PGE2 secreted locally acts as a secretagogue factor and causes electrolytes’ loss [11,12,13]. The use of COX inhibitors to control volume loss should only be considered as a temporary measure until the patient is stable enough for definitive surgical resection. The treatment’s cornerstone is surgery, to remove villous adenoma after fluid and electrolyte imbalances have been corrected [5,6]. Left untreated, the mortality rate of secreting villous adenoma is 100% [14].

The particularity of our case is that our patient presented with clinical and paraclinical features suggestive of McKittrick-Wheelock syndrome and had associated *Clostridioides difficile* infection. We found only one case-report published until now regarding the link between *Clostridioides difficile* infection and McKittrick and Wheelock syndrome [1]. Learney et al. [2] described in their paper the case of a patient that had two distinct episodes of diarrhea associated with acute renal failure resulting from two independent colo-rectal conditions: first *Clostridioides difficile* infection and then McKittrick-Wheelock syndrome. It has been reported that patients presenting with bowel obstruction have the highest incidence of *Clostridioides difficile* infection, around 29% [9]. Colon obstruction, even incomplete, by the rectal tumor determines bowel clearance reduction. As a result, dysbiosis appears. This could explain the community acquired *Clostridioides difficile* infection and its persistence despite optimal therapy in our patient. Community acquired *Clostridioides difficile* infection might be considered secondary to the alteration in the colonic microenvironment due to rectal obstruction. In addition, our patient had a history of complicated diabetes, and it is well known that community acquired *Clostridioides difficile* infection is more likely to occur in elderly patients with certain co-morbidities and risk factors [2].

## 4. Conclusions

McKittrick-Wheelock syndrome is a rare disease with high mortality if it is not properly treated, and which can predispose to community acquired *Clostridioides difficile* infection. This case-report highlighted the link between two independent colo-rectal diseases responsible for watery diarrhea and acute renal failure.

## Figures and Tables

**Figure 1 diagnostics-12-00784-f001:**
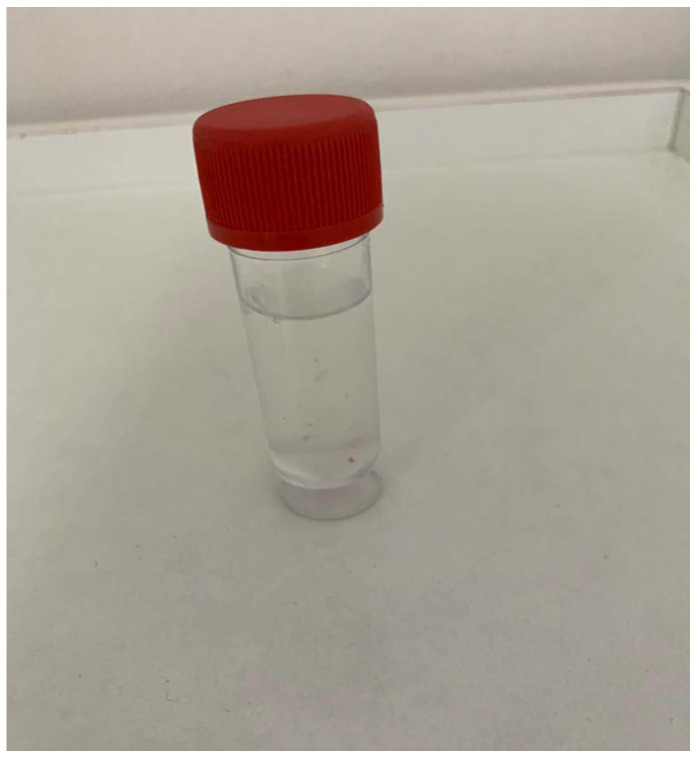
Patient stool’s aspect.

**Figure 2 diagnostics-12-00784-f002:**
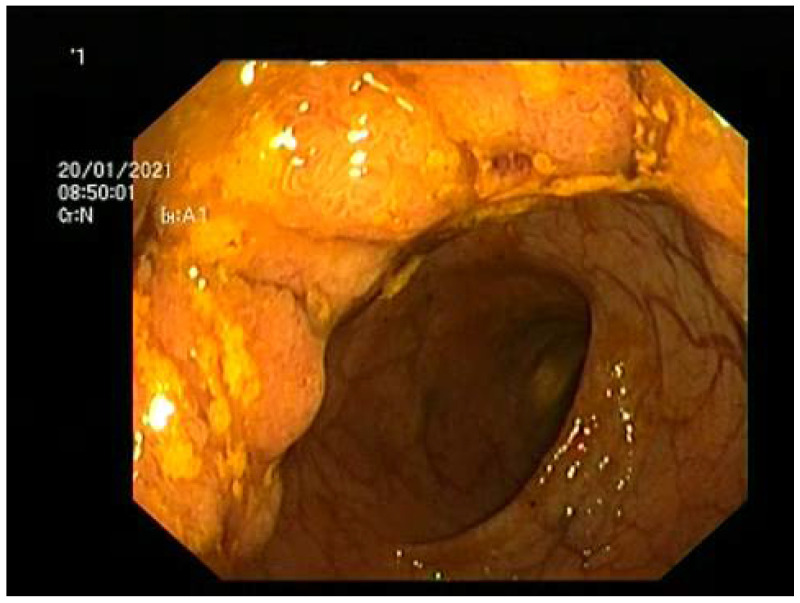
Tumoral mass identified at colonoscopy.

**Figure 3 diagnostics-12-00784-f003:**
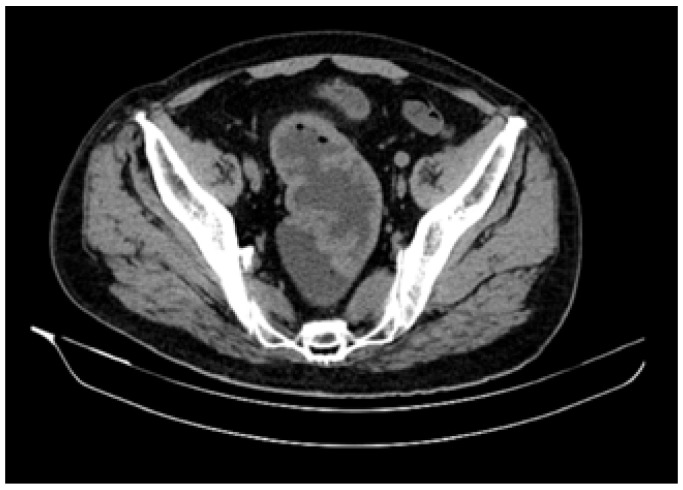
Computer tomography features—a 12 cm in length and 2 cm in thickness recto-sigmoid circumferential tumor.

**Figure 4 diagnostics-12-00784-f004:**
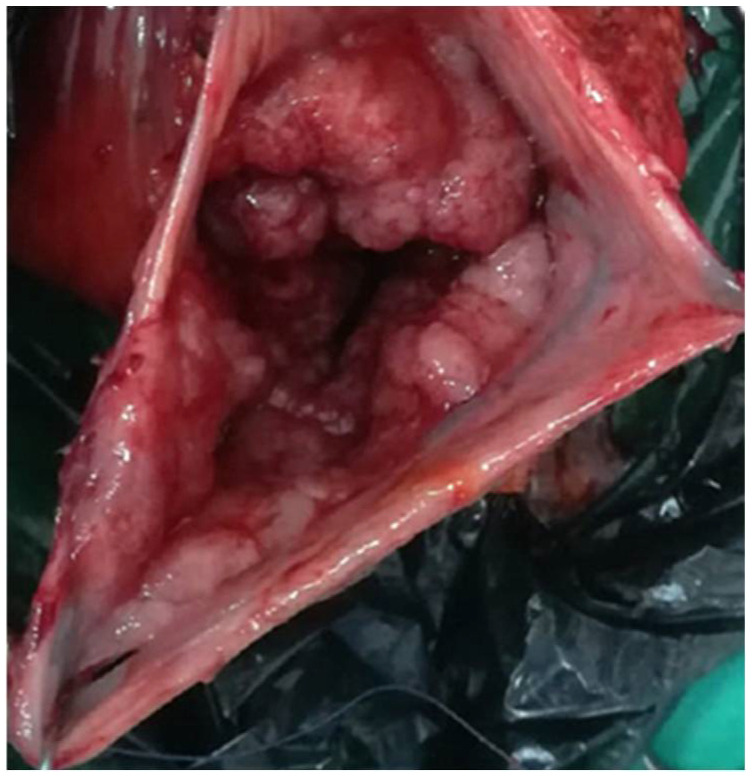
Resected specimen.

**Figure 5 diagnostics-12-00784-f005:**
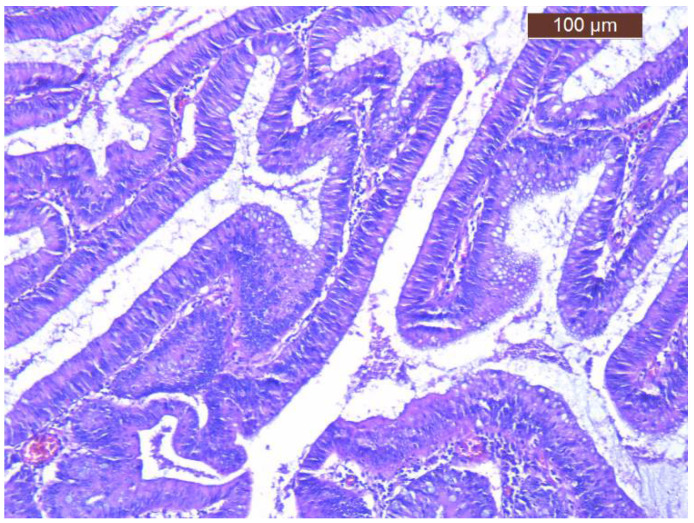
Histology of resected specimen (tubulo-villous adenoma with low-grade dysplasia and several foci of high-grade dysplasia).

**Table 1 diagnostics-12-00784-t001:** Blood tests results.

Parameter	Normal Reference Range	Initial Presentation	Admission in Gastroenterology Department	Week 1	Week 2	1 Week after Surgery
Sodium (mmol/L)	136–145	130	135	140	140	142
Potassium (mmol/L)	3.5–5.1	2.8	3.4	2.5	3	4.1
Chloride (mmol/L)	98–107	85	87	99	107	112
Bicarbonate (mmol/L)	23–31	14	24.5	26.6	20.7	22
Urea (mg/dL)	18–55	300	110	171	41	69
Creatinine (mg/dL)	0.72–1.25	11	1.4	2.27	1.15	1.21

**Table 2 diagnostics-12-00784-t002:** Stool tests results.

Parameter	Initial Presentation	Admission in Gastroenterology Department	Week 1	Week 2	1 Week after Surgery
*Clostridioides difficile* infection (Toxin A and B by direct enzyme immunoassay)	+	+	+	+	-

## Data Availability

Not applicable.

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
