# Peer review of "Uncommon Association of Mckittrick-Wheelock Syndrome and Clostridioides difficile Infection in Acute Renal Failure"

_diagnostics, 2022, doi:10.3390/diagnostics12040784_

Round 1
Reviewer 1 Report
Comments to the Author
Thank you for allowing me to review your manuscript titled “UNCOMMON ASSOCIATION OF McKITTRICK-WHEELOCK SYNDROME AND CLOSTRIDIOIDES DIFFICILE INFECTION IN ACUTE RENAL FAILURE”.
The authors have reported a case of acute renal failure associated with Clostridioides difficile infection and McKittrick-Wheelock syndrome.
My comments are as follows:
- I speculate that the patient with McKittrick-Wheelock syndrome presented worsening of secretory diarrhea due to the Clostridioides difficile The authors should discuss the relationship between McKittrick-Wheelock syndrome and the Clostridioides difficile infection.
- In the Conclusions section, the authors have stated that Clostridioides difficile infection was maintained by the presence of tumoral obstruction. However, the endoscopic image (Figure 1) reveals preserved intestinal lumen with no tumoral obstruction. The authors should find an appropriate rationale to discuss why Clostridioides difficile infection was maintained in the patient.
- In the Introduction section, the authors have stated that the patient presented contact with a healthcare facility 12 weeks prior, while in the Case presentation section, they have stated that the patient denied any contact with a healthcare facility in the previous 12 weeks. Please explain these contradictory statements.
- In the colonoscopic image, the tumor was reported as a 10 cm length circumferential lesion located 8 cm from the anal verge, while in the CT scan the tumor was illustrated as 12 cm in length located 13 cm from the anal verge. These statements would confuse the readers. Please correct it.
- In the endoscopic image (Figure 1), the thick mucus adheres to the surface of the tumor, making it difficult to visualize the appearance of the tumor. Please select an appropriate image.
- In Figure 4, please add a histological macrograph showing the villous structure of the tumor.
Author Response
- Complete colon obstruction by the rectal tumor determines reduction of bowel clearance. As a result, dysbiosis appears. Could explain the community-acquired Clostridioides difficile infection and its persistence.
- Thank you for the observation. We will change the endoscopic image (Figure 1) with even more relevant one. The colon obstruction was incomplete, which makes the bowel evacuation more difficult.
- Thank you again for your relevant observation. By mistake, we omitted the word “without” in the “Introduction” section. Our patient din not have any contact with a healthcare facility in the previous 12 weeks.
- We reported two different measurements as we were talking about two different diagnostic methods: CT and colonoscopy. We appreciate your comment. The statements might be confusing for readers, so we will guide after the CT measurements.
- You are absolutely right. The endoscopic image (Figure 1) makes difficult to visualize the appearance of the tumor, so we will select another image.
- We will add a histological macrograph showing the villous structure of the tumor. Thank for your kind suggestion.
Reviewer 2 Report
The case report entitled UNCOMMON ASSOCIATION OF McKITTRICK-WHEE-2 LOCK SYNDROME AND CLOSTRIDIOIDES DIFFICILE INFECTION IN ACUTE RENAL FAILURE is very interesting, and the presentation is clear, the scientific quality of the paper is good.
I recommend some minor revisions of the paper:
- According to guidelines ,a recurrent C difficile infection is defined as the recurrence of diarrhea and a confirmatory positive test (NAAT or EIA) .
- The sentence from the Discussions: "There have been no reports that would link Clostridioides difficile infection with McKittrick and Wheelock syndrome" should be changed to: " There is one similar case report in the literature with an association of Clostridioides difficile infection with McKittrick and Wheelock syndrome " and the differences and similarities should be discussed: (Learney RM, Ziprin P, Swift PA, Faiz OD. Acute Renal Failure in Association with Community-Acquired Clostridium difficile Infection and McKittrick-Wheelock Syndrome. Case Rep Gastroenterol. 2011 May;5(2):438-44. doi: 10.1159/000330478. Epub 2011 Aug 18. PMID: 21960946; PMCID: PMC3180660)
- Some Romanian Case Report can be cited:
-
Mois EI, Graur F, Sechel R, Al-Hajjar N. McKittrick-Wheelock syndrome: a rare case report of acute renal failure. Clujul Med. 2016;89(2):301-303. doi:10.15386/cjmed-536
- Preda CM, Meianu C, Becheanu G, Dumbrava M, Manuc M, Diculescu M. Fecal Microbiota Transplantation in Recurrent Nap1/B1/027 Clostridium Difficile infection CDI) resistant to Vancomycin and Metronidazole in a patient with ulcerative colitis (UC): A case report. Rev Med Chir Soc Med Nat 2016 120 (3): 563-567.
-
Author Response
- Thank you for your kind observation. You are perfectly right regarding the definition of recurrent Clostridioides difficile (CD) infection, but we consider resistance to-first line therapy wwith Vancomycin and Metronidazole. Therefore, second-line therapy with Fidaxomicin was administered. In fact, we think that the lack of response to treatment was due to both rectal tumor and CD infection.
- We appreciate your observation. We will change the sentence as you kindly indicated. The case-report of Learney et al that you mentioned refers to a patient that had two clinical conditions responsible of diarrhea, but not at the same time (first CD infection, than McKittrick Wheelock syndrome).
- Thank you for indicating the references. We will cite it in our manuscript.

Reviewer 3 Report
manuscript entitled "Uncommon Association of McKittrick-Wheelock Syndrome and Clostridioides difficile Infection in Acute Renal Failure"
Major issues:
- The association between this conditions had been reported previously and lacks novelty.
- The description of the case is not delicate and informative enough. The authors must provide lab date and histological findings in a more detail manner.
- The discussion is not informative. The association has not been discussed. The authors should provide a detail review on published cases to gain more insights.
Author Response
- There are several cases of McKittrick-Wheelock syndrome published until now, but we found only one case of McKittrick-Wheelock syndrome that also had Clostridioides difficile infection, but not simultaneously. That is the reason why we considered useful reporting this case.
- Thank you for the observation. We considered the lab data easier to follow if they were presented in a table. Regarding the histological findings, we will provide a more detailed description.
- Thank you again for your relevant comment. We will include in the “Discussion” section details about the impact of rectal tumor on patients with Clostridioides difficile infection and we will discuss this association with regard to the previous studies published.
Reviewer 4 Report
This is a clear case report of a large rectal adenoma associated with C difficile infection. The authors have not discussed whether there is a biologically plausible mechanism for the association (perhaps via alteration in the colonic microenvironment) and I would recommend they do so. They also have not listed all relevant risk factors for C difficile - the patient's medication history should be briefly summarised. Also the method of C diff toxin testing should be described - if it was a PCR test, persistent positivity is to be expected, and this should be discussed.
Minor suggestions for improving the manuscript follow:
Page 2 line 50 suggest "he was treated with" instead of "there were administered"
Page 2 line 58, suggest "Because of this persisting symptom" instead of "Given all these"
Page 2, line 59 suggest "investigation" instead of "follow-up"
Page 2 line 69 describe the extent of colonoscopy
Page 4 line 110 Suggest "reported" instead of "presented"
Overall the case is well described and of sufficient interest for publication.
Author Response
Thank you for your review and for your kind recommendations. We will provide a more detailed discussion about the mechanism underlying the association between Clostridioides difficile infection and McKittrick-Wheelock syndrome. Indeed, we considered the alteration of colonic microenvironment due to incomplete obstruction by rectal tumor.
Regarding the risk factors for Clostridioides difficile infection and the patient's medication history, we mentioned in the text that our patient had a medical history of diabetes with microangiopathic complications, treated with insulin. He had no history of other medication.
For the diagnosis of Clostridioides difficile infection, detection of toxin A/B in stool specimens by direct enzyme immunoassay was used. Under the circumstances of lack of response of diarrhea after two lines of treatment and under conditions of the surgery department, retesting for Clostridioides difficile infection was done in order to distinguish between three possible situation: Clostridioides difficile infection, Clostridioides difficile infection and rectal tumor or diarrhea due to the rectal tumor. We had to proceed this way, despite the fact that current guidelines do not recommend retesting after treatment.
Thank you also very much for English corrections.
Round 2
Reviewer 1 Report
Thank you for allowing me to re-review your manuscript titled “Uncommon Association of McKittrick-Wheelock Syndrome and Clostridioides difficile Infection in Acute Renal Failure”. The revised manuscript is well organized.
Reviewer 3 Report
This is not the first case. Similar report has been published in "Case Rep Gastroenterol 2011 May;5(2):438-44, .... and more. I encourage the authors to make a more detail literature search.